# The Formation of Magnesite Ores by Reactivation of Dunite Channels as a Key to Their Spatial Association to Chromite Ores in Ophiolites: An Example from Northern Evia, Greece

Giovanni Grieco [1], Alessandro Cavallo [2], Pietro Marescotti [3], Laura Crispini [3], Evangelos Tzamos [4] and Micol Bussolesi [2,*]

1 Department of Earth Sciences, University of Milan, Via Botticelli 23, 20122 Milan, Italy
2 Department of Earth and Environmental Sciences—DISAT, University of Milan-Bicocca, P.zza della Scienza 1-4, 20126 Milan, Italy
3 Department of Earth Environment and Life Sciences—DISTAV, University of Genoa, Corso Europa 26, 16126 Genoa, Italy
4 Ecoresources PC, Giannitson and Santarosa Str. 15-17, 54627 Thessaloniki, Greece
* Correspondence: micol.bussolesi@unimib.it

**Abstract:** Ophiolite magnesite deposits are among the main sources of magnesite, a raw material critical for the EU. The present work focuses on magnesite occurrences at Kymasi (Evia Island, Greece), in close spatial association with chromitite within the same peridotite massif, and on the relationship between ultramafic rocks and late magnesite veins. Chromitite lenses are hosted within dunite, in contact with a partially serpentinized peridotite cut by magnesite veins. Close to the veins, the peridotite shows evidence of carbonation (forming dolomitized peridotite) and brecciation (forming a serpentinite–magnesite hydraulic breccia, in contact with the magnesite veins). Spinel mineral chemistry proved to be crucial for understanding the relationships between different lithologies. Spinels within partially serpentinized peridotite (Cr# 0.55–0.62) are similar to spinels within dolomitized peridotite (Cr# 0.58–0.66). Spinels within serpentinite–magnesite hydraulic breccia (Cr# 0.83–0.86) are comparable to spinels within dunite and chromitite (Cr# 0.79–0.84). This suggests that older weak zones, such as dunite channels, were reactivated as fluid pathways for the precipitation of magnesite. Magnesite stable isotope composition, moreover, points towards a meteoric origin of the oxygen, and to an organic source of carbon. The acquired data suggest the following evolution of Kymasi ultramafic rocks: (i) percolation of Cr-bearing melts in a supra-subduction mantle wedge within dunite channels; (ii) obduction of the ophiolitic sequence and peridotite serpentinization; (iii) uplift and erosion of mantle rocks to a shallow crustal level; (iv) percolation of carbon-rich meteoric waters rich at shallow depth, reactivating the dunite channels as preferential weak zones; and (v) precipitation of magnesite in veins and partial brecciation and carbonation of the peridotite host rock.

**Keywords:** magnesite; chromite; critical raw materials; ophiolite; serpentinite

## 1. Introduction

Magnesium (Mg) is a Critical Raw Material (CRM) for the EU, important for the automotive industry, desulphurization of steel, packaging applications, and construction equipment, as well as some other minor uses [1]. Magnesite ore deposits are the main source of Mg for the industry and are used in the production of various kinds of magnesium compounds. They occur in two main exploitable deposit types: cryptocrystalline magnesite, hosted by ultramafic rocks ("Kraubath type"), and sparry magnesites, hosted by marine platform carbonates ("Veitsch type") [2–4]. While Veitsch-type deposits are more common and have a larger size, Kraubath-type ones usually contain higher-quality magnesite, and are hence preferred for exploitation. Kraubath-type cryptocrystalline magnesite ores usually

form irregular bodies frequently associated with the ultramafic rocks of ophiolite suites. The sources of Mg are the Mg-rich minerals of ultramafic rocks, in particular peridotites undergoing serpentinization processes [5]. $CO_2$, on the other hand, can be ascribed to various sources, including organic ones, carbonate rocks, meteoric waters, or hypogenic hydrothermal fluids [6–8].

Magnesite deposits, moreover, are a natural case study that can provide useful information for the well-known issue of $CO_2$ sequestration. The study of naturally occurring, ultramafic-hosted magnesites provides a useful playground for understanding the mechanisms of mineral carbonation, both in fully carbonated peridotites and in partially carbonated ones [9].

Greece hosts exploitable ophiolite-related magnesite deposits in two areas: the Evia Island and the Chalkidiki peninsula [10,11]. The ore in Northern Evia consists of a shallow (200 m) stockwork of fracture-filling magnesite veins [12], hosted within peridotites (harzburgites and lherzolites) with various degrees of serpentinization.

Kraubath-type magnesite deposits, due to their geological context, can be spatially associated with other ophiolite-related ore deposits, for example, in the present study, to chrome ores. A similar relation can be found in the Gerakini area (western Chalkidiki), where currently exploited magnesite deposits are found closely associated with chromitites.

Evia Island hosts active magnesite mines operated by TERNA MAG and Grecian Magnesite Mining companies. Several, never-exploited chromitite occurrences have been described in Northern Evia not far from the major magnesite deposits.

This work is focused on minor magnesite occurrences at Kymasi (Evia), that were selected for their very good exposure on an easily accessible horizontal outcrop by the sea, and for their very close association with nearby outcropping chromitite occurrences within the same peridotite massif, hence providing an optimal playground for studying the relationship between two different ores. In this area, magnesite was once extracted in an open pit, and the ore was carried to the nearby port to be transported by sea.

In the present study, we investigate the origin of the spatial relationship between magnesite and chromite deposits. Chromium and magnesite are two georesources considered (chromium) or listed (magnesium) as Critical Raw Materials for the EU. This mineralogical association is hosted within peridotites, the main source of olivine, and can therefore provide an interesting case of sustainable mining. The close spatial relationship between these three commodities could be useful for a reduced-waste, more sustainable exploitation.

## 2. Geological Setting

Evia is located in central Greece, and it is the second largest island of the country. The island is part of the Hellenides belt, which is divided into two tectonic units: the Internal and External Hellenides. Evia is geologically comprised in the southern part of the Internal Hellenides (Figure 1a), within the Pelagonian Zone (Northern Evia) and the Atticocycladic Zone (Southern Evia). These are two of the many geotectonic domains in which the Internal Hellenides are subdivided. The Atticocycladic Zone exposes a pile of high-pressure tectonic nappes overlain by the Pelagonian Zone [13–15]. The Pelagonian Zone is formed by an assemblage of tectonic units mainly consisting of a Paleozoic metamorphic basement [13,16]. The crystalline basement is overthrust by Late Jurassic ophiolites and Upper Paleozoic and Mesozoic sedimentary sequences [15,17].

The mafic and ultramafic rocks of Evia (Figure 1b) are generally regarded as belonging to the "Western Ophiolite Belt", along with Vourinos, Pindos, and Othris ophiolites, in contrast with the "Eastern Ophiolite Belt", comprising the Vardar Zone and Guevgueli ophiolites, with chromitite occurrences at Vavdos, Gerakini-Ormilya, Vasilika, and Krani [18], and the chromitite occurrences of the Serbo-Macedonian Massif. The ultramafic rocks in the area are known for hosting large magnesite deposits, which are currently exploited [19]. Moreover, chromitite occurrences are reported in the literature in three localities, Papades, Mandoudi, and Limni [20], in small outcrops associated with serpentinites.

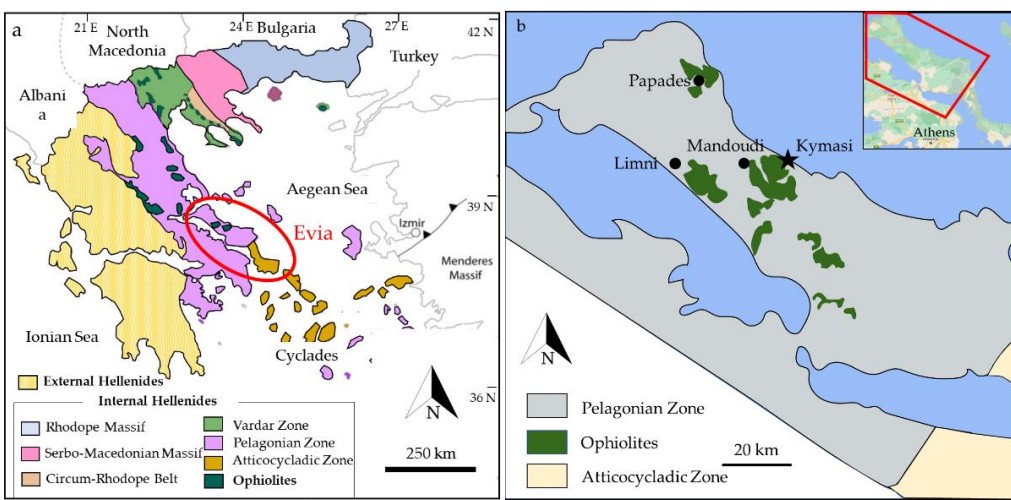

**Figure 1.** (**a**) Simplified map showing the major structural elements of Greece. Modified after [21,22]; (**b**) Simplified map of Norther Evia showing the major ophiolite bodies of the area. Kymasi mineralization are indicated by the black star. Modified after [23].

## 3. Materials and Methods

At Kymasi, field work was focused on a well-exposed network of peridotite-hosted magnesite veins (38°48′28.37″ N, 23°31′13.32″ E) and on small chromitite lenses cropping out at a distance of about 120 m (38°48′25.24″ N, 23°31′16.98″ E) (Figure 2).

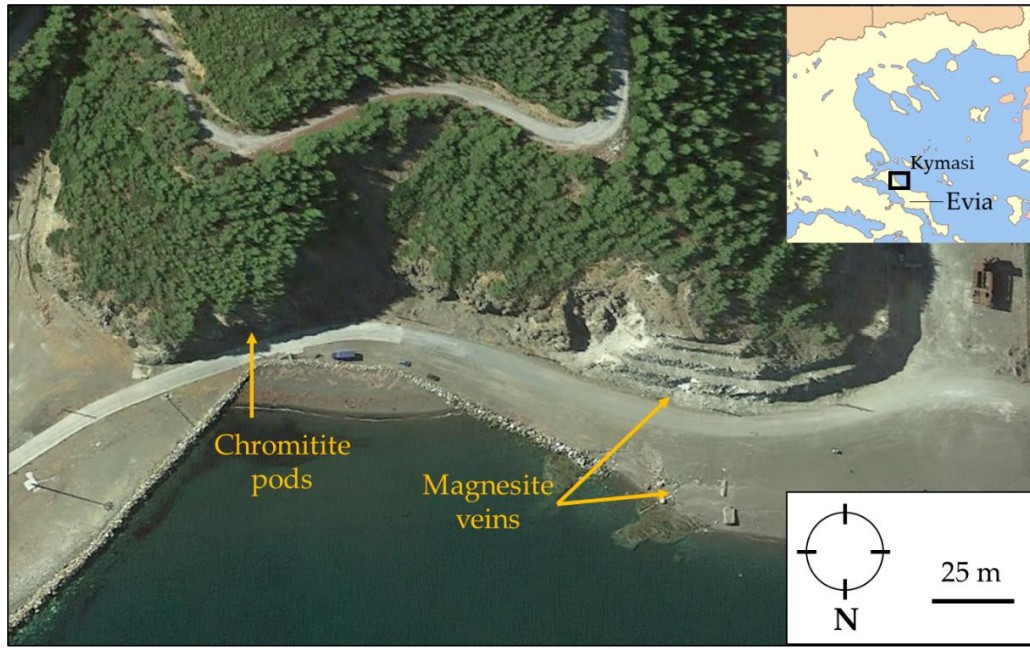

**Figure 2.** Sampling position (Google Earth®) of chromitite pods, magnesite veins, and their host rock.

A centimetric-to-decimetric chromitite lens at Kymasi is hosted within partially altered dunite, in contact with a harzburgite host rock cut by magnesite veins (Figure 3). Chromitites are densely disseminated, with ~50% chromite modal content. Five chromitite and two dunite specimens were sampled from the outcrop.

Magnesite veins occur as persistent, metric conjugate vein systems (Figure 4a) with orientation NE–SW, NW–SE, and E–W, and with variable thickness, from a few cm up to 50 cm. The center of the veins is composed of white massive magnesite, while at the rim, magnesite shows a reddish secondary alteration (Figure 4b,c).

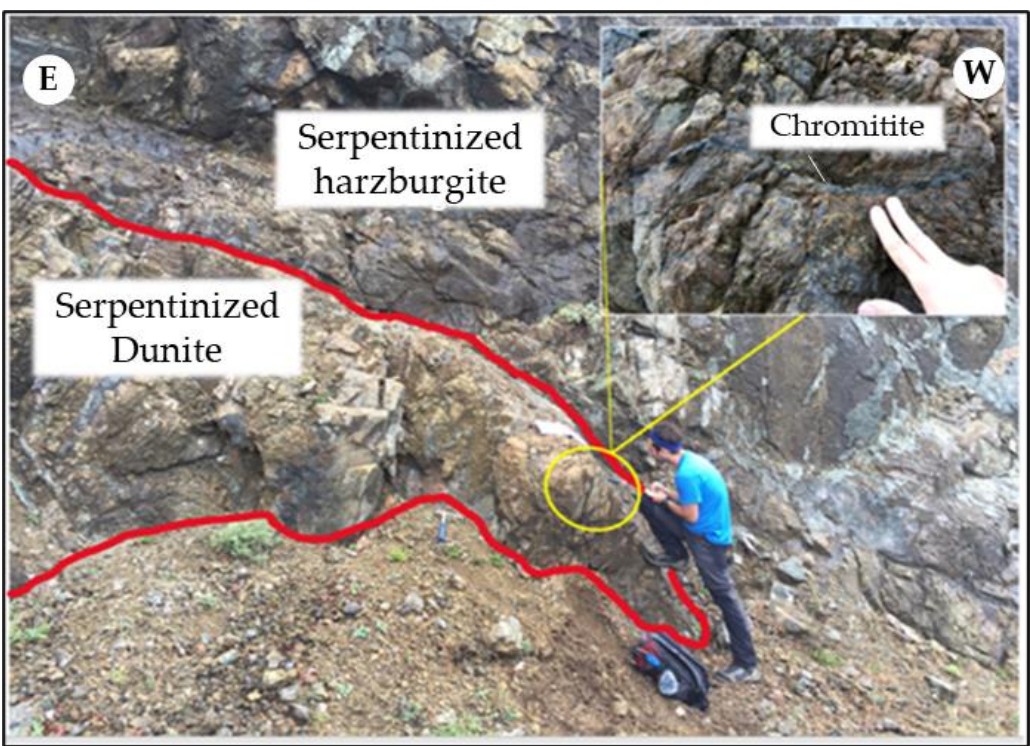

**Figure 3.** Chromitite pods in serpentinized dunite, hosted by partially serpentinized harzburgite at Kymasi.

Three massive magnesite specimens were sampled from the cores of 3 vein sets (F1, F2, and F3). Vein F2 was then sampled in more detail. From core to rim, the collected samples are: a thick massive magnesite central portion (F4a), a serpentinite breccia with magnesite cement (F4b), and a partially dolomitized peridotite (F4c). The host rock is a partially serpentinized peridotite (F4d). All contacts are irregular and sharp (Figure 4b,c).

The collected samples were studied through optical microscopy in transmitted and reflected light, X-ray powder diffraction (XRPD), scanning electron microscope (SEM-EDS), and electron probe microanalyzer (EMPA). Magnesite samples were also analyzed for their stable carbon and oxygen isotopes.

XRPD analyses were performed at the University of Milan with a PANalytical X'Pert-pro instrument set at the following operating conditions: 40 kV of voltage; 40 mA of current; and Cu anticathode K$\alpha$1/K$\alpha$2: 1.540510/1.544330 Å. This powder diffractometer is equipped with an incident beam monochromator, which separates the K1 and the K2 and works with the Bragg–Brentano geometry. Analyses were conducted with a step size [°2Th.] of 0.0170 (scan time 30.3607 s). Data were elaborated with software X'Pert Highscore v.2.3.

$\mu$-Raman analyses were performed at the University of Genova with the HORIBA XploRA_Plus Raman Spectrometer (532 nm laser line), equipped with the OLYMPUS BX-41 optical microscope. Raman spectra were acquired using a 2400 grooves mm$^{-1}$, with an acquisition time of 100 s.

Preliminary SEM analyses were conducted at the University of Milano-Bicocca using a Tescan VEGA TS 5136XM equipped with an EDS analyzer (EDAX Genesis 400), with 200 pA and 20 kV and high-vacuum as standard conditions, on carbon-coated samples (20 nm) using an Edwards 5150B carbon coater.

Mineral chemistry was determined through a JEOL 8200 electron microprobe (JEOL Ltd., Akishima, Japan) equipped with a wavelength dispersive system (WDS) at the Earth Sciences Department of the University of Milan. The microprobe system operated using an accelerating voltage of 15 kV, a sample current on brass of 15 nA, and a counting time of 20 s on the peaks and 10 s on the background. Beam diameter is 1 $\mu$m. A series of natural minerals was used as standards: wollastonite for Si, forsterite for Mg, ilmenite

for Ti, fayalite for Fe, anorthite for Al and Ca, metallic Cr for Cr, niccolite for Ni, and rhodonite for Mn and Zn. Detection limit is approximately 330 ppm for Ti (6% standard deviation), 460 ppm for Mn (6% standard deviation), 160 ppm for Mg (2% standard deviation), 180 ppm for Si (6% standard deviation), 320 ppm for V (12% standard deviation), 370 ppm for Fe (8% standard deviation), 140 ppm for Ca (15% standard deviation), 135 ppm for Al (4% standard deviation), 370 ppm for Cr (10% standard deviation), 390 ppm for Ni (9%standard deviation), and 800 ppm for Zn (9% standard deviation). Analyses were conducted on carbon-coated samples (20 nm) using an Edwards E306A coating device.

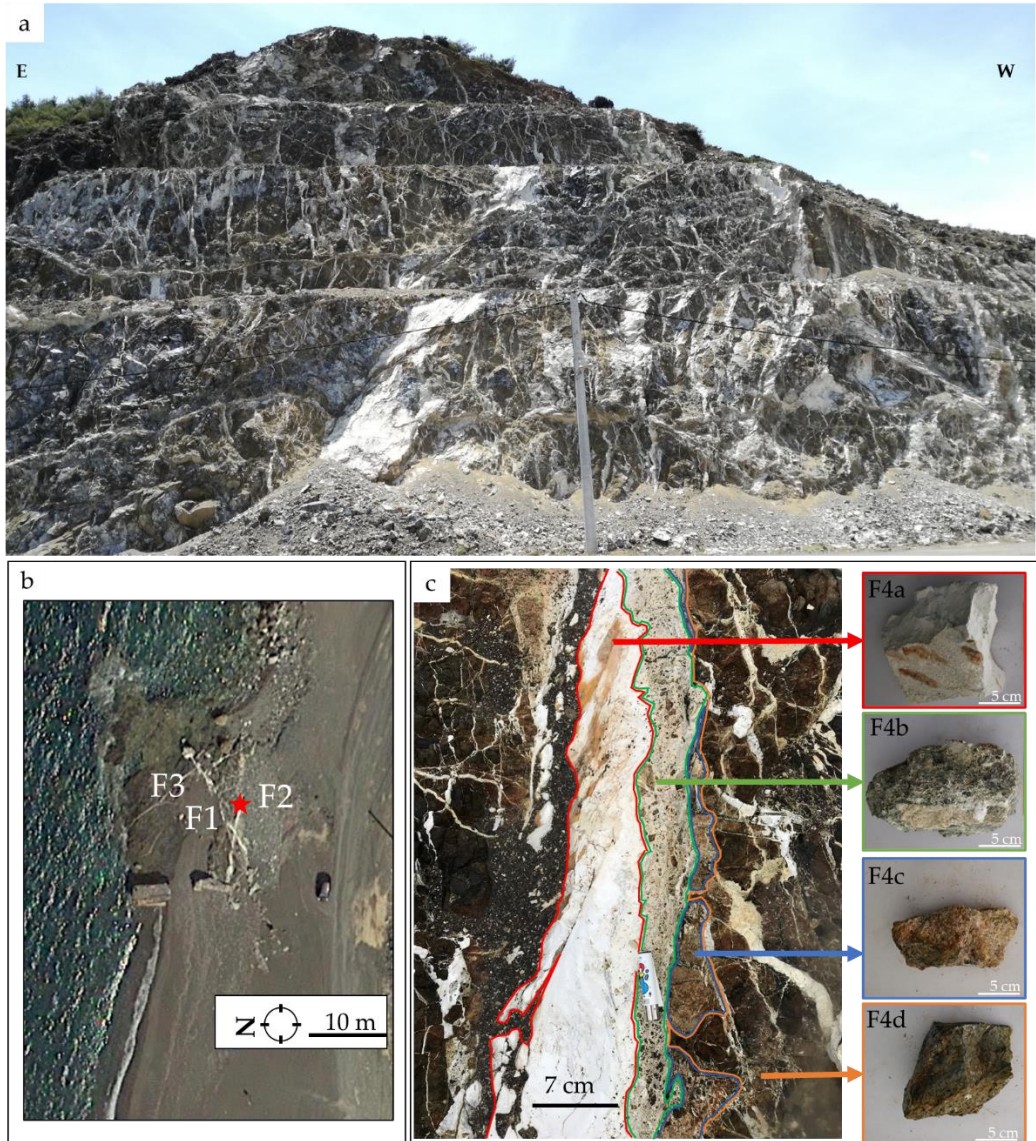

**Figure 4.** (**a**) Magnesite vein network hosted within variably altered peridotite; (**b**) Sampled magnesite veins (red star), and (**c**) detail of the magnesite vein F2 and sampled host lithologies.

Stable isotope analyses were performed at the University of Milan using a Delta V Advantage mass spectrometer coupled with a GasBench II. Phosphoric acid is added to the solid sample, and the resulting $CO_2$ is then analyzed.

## 4. Results

### 4.1. Mineralogy and Texture

The mineralogy of chromitite and dunite, magnesite veins (sample F1, F2, F3, and F4a), serpentinite–magnesite breccia (sample F4b), dolomitized peridotite (sample F4c),

and partially serpentinized peridotite (sample F4d) has been analyzed by means of optical microscopy, XRPD, and μ-Raman.

### 4.1.1. Chromitite and Dunite

Kymasi densely disseminated chromitites consist of a 40–50 cm-thick schlieren body made up of thin layers of chromitite ~2 cm thick, containing up to 50% chromite, alternated to thin dunite layers up to 10 cm thick. Chromite crystals are submillimetric (50–500 μm), and show rare alteration into ferrian chromite. The interstitial silicate matrix is composed of serpentine, forming a mesh texture, with olivine relicts and rare chlorites associated with ferrian chromite (Figure 5a). Primary phases in dunite are mainly olivine relicts partially replaced by lizardite and chrysotile along the rims (Figure 5b). Cr-spinels are present as accessory phases.

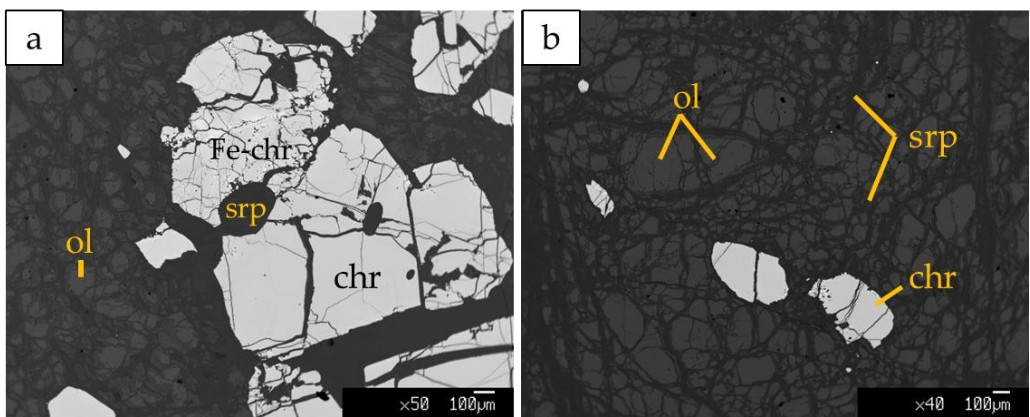

**Figure 5.** BSE images of (**a**) densely disseminated chromitites at Kymasi; (**b**) Dunite with accessory spinel in contact with chromitites. Fe-chr: ferrian chromite, ol: olivine, chr: chromite, srp: serpentine.

### 4.1.2. Carbonate-Bearing Peridotites and Magnesite Mineralization

The host rock of magnesite mineralization is a partially serpentinized peridotite (sample F4d), composed of olivine partially replaced by lizardite, minor tremolite, orthopyroxene, clinopyroxene, and accessory spinel. No carbonate minerals were detected (Figure 6a). Chrysotile occurs in very subordinate amounts within sub-millimetric veinlets as well as within rims of mesh texture or cleavage planes of bastites (Figure 7a,b). Chrysotile is often associated with talc and chlorite.

The massive magnesite veins are characterized by secondary halos at the contact with the host peridotite composed of dolomitized peridotite (sample F4c; Figure 6b,c) and a serpentinite–magnesite hydraulic breccia (sample F4b, Figure 6d) moving towards the massive magnesite.

The dolomitized peridotite is mainly composed of dolomite crystals with a euhedral-to-subhedral habit, lizardite, and subordinate clinopyroxene and orthopyroxene relicts (Figures 6b,c and 7a). Spinel is present as an accessory phase.

The serpentinite–magnesite hydraulic breccia consists of poorly sorted angular clasts of chromite-rich serpentinite breccia cemented by magnesite with a cockade texture (Figures 6d and 7a). Lizardite is the main constituent of serpentinite clasts with minor amounts of chrysotile, talc, and chlorite occurring within sub-millimetric veinlets and along rims of mesh texture. The cement of the breccia is composed by cryptocrstalline magnesite with massive texture and, locally, by magnesite and minor dolomite with a microcrystalline polygonal texture (Figure 7b).

Massive magnesite, sampled from the core of the veins, is composed mainly of cryptocrystalline magnesite Figures 6e and 7a), with minor sparitic and spherulitic magnesite. The other mineralogical phases are serpentine relicts and microfracture-filling talc (Figure 6f).

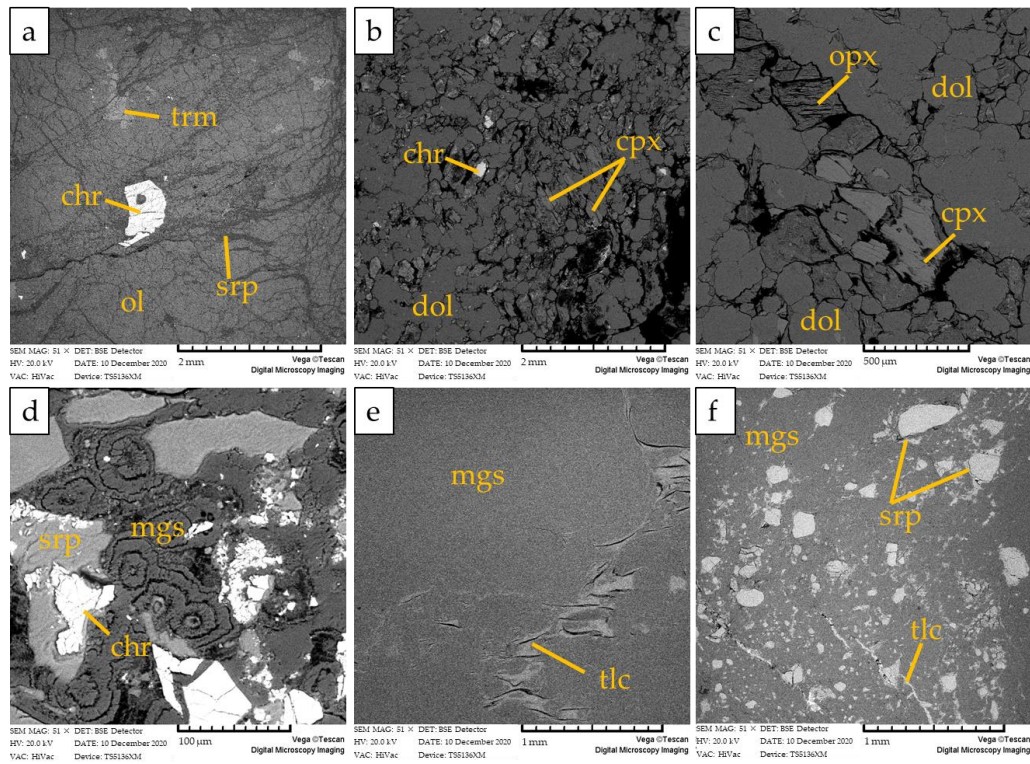

**Figure 6.** BSE images of (**a**) partially serpentinized peridotite (F4d), showing accessory spinel associated with unaltered olivine, rarely altered into serpentine; (**b**) dolomitized peridotite (F4c) showing euhedral dolomite crystals and rare spinel and clinopyroxene; (**c**) dolomitized peridotite showing euhedral dolomite crystals and subordinate clinopyroxene and orthopyroxene; (**d**) hydraulic breccia (F4b) showing angular serpentine and chromite clasts cemented by magnesite with cockade texture; (**e**,**f**) massive magnesite (F4a) with minor talc and serpentine clasts. Mineral abbreviations are trm: tremolite, chr: chromite; ol: olivine; srp: serpentine; cpx: clinopyroxene; dol: dolomite; opx: orthopyroxene; mgs: magnesite; tlc: talc.

### 4.2. Mineral Chemistry

The composition of spinels within chromitites, partially serpentinized peridotite, partially dolomitized peridotites, and serpentinite–magnesite hydraulic breccia is reported in Table 1, Figure 8a, and in the Supplementary Materials. The composition of olivine, pyroxene, and carbonates is reported in Figure 8b–d and in the Supplementary Materials. Complete chemical analyses of serpentine, chlorite, and amphibole are reported in the Supplementary Materials.

**Table 1.** Major and minor elements average composition and standard deviation of chromite cores from Kymasi lithologies; Mg# = [Mg/(Mg + Fe$^{2+}$)]; Cr# = [Cr/(Cr + Al)]. Fe$_2$O$_3$ and FeO are calculated based on stoichiometry.

| | Chromitite | | Chromitite | | Partially Serp. Peridotite (F4d) | | Dolomitized Peridotite (F4c) | | Serpentinite–Mag. Breccia (F4b) | |
|---|---|---|---|---|---|---|---|---|---|---|
| | Unaltered Chromite n = 51 | | Ferrian Chromite n = 41 | | Chromite n = 15 | | Chromite n = 5 | | Chromite n = 5 | |
| Wt% | avg | st.dev. | avg | st.dev. | avg | st.dev. | avg | st.dev. | avg | st.dev. |
| TiO$_2$ | 0.10 | 0.04 | 0.08 | 0.04 | 0.11 | 0.06 | 0.06 | 0.06 | 0.06 | 0.04 |
| Al$_2$O$_3$ | 8.47 | 0.74 | 4.30 | 1.82 | 24.14 | 2.67 | 18.62 | 1.71 | 7.96 | 2.76 |
| Cr$_2$O$_3$ | 62.84 | 1.58 | 64.96 | 2.12 | 44.83 | 4.27 | 50.17 | 1.62 | 61.14 | 3.84 |
| Fe$_2$O$_3$ | 2.02 | 0.75 | 2.16 | 1.05 | 2.13 | 1.78 | 1.64 | 0.98 | 1.76 | 1.78 |

**Table 1.** *Cont.*

| | Chromitite | | Chromitite | | Partially Serp. Peridotite (F4d) | | Dolomitized Peridotite (F4c) | | Serpentinite–Mag. Breccia (F4b) | |
|---|---|---|---|---|---|---|---|---|---|---|
| | Unaltered Chromite n = 51 | | Ferrian Chromite n = 41 | | Chromite n = 15 | | Chromite n = 5 | | Chromite n = 5 | |
| FeO | 15.42 | 1.44 | 21.36 | 1.09 | 12.86 | 0.55 | 17.31 | 1.07 | 18.55 | 3.96 |
| MnO | 0.12 | 0.14 | 0.31 | 0.10 | 0.07 | 0.15 | 0.19 | 0.17 | 0.27 | 0.24 |
| MgO | 11.92 | 0.91 | 7.41 | 0.92 | 14.51 | 0.97 | 11.30 | 0.49 | 9.22 | 2.73 |
| Total | 100.93 | 1.00 | 100.62 | 0.80 | 98.90 | 0.56 | 99.36 | 0.67 | 99.24 | 1.08 |
| Ti | 0.00 | 0.00 | 0.00 | 0.00 | 0.00 | 0.00 | 0.00 | 0.00 | 0.00 | 0.00 |
| Al | 0.33 | 0.03 | 0.17 | 0.07 | 0.87 | 0.08 | 0.70 | 0.06 | 0.31 | 0.10 |
| Cr | 1.62 | 0.04 | 1.77 | 0.09 | 1.08 | 0.12 | 1.26 | 0.05 | 1.64 | 0.14 |
| $Fe^{3+}$ | 0.05 | 0.02 | 0.06 | 0.03 | 0.05 | 0.04 | 0.04 | 0.02 | 0.04 | 0.04 |
| $Fe^{2+}$ | 0.42 | 0.04 | 0.61 | 0.04 | 0.33 | 0.02 | 0.46 | 0.03 | 0.53 | 0.13 |
| Mn | 0.00 | 0.00 | 0.01 | 0.00 | 0.00 | 0.00 | 0.00 | 0.01 | 0.01 | 0.01 |
| Mg | 0.58 | 0.04 | 0.38 | 0.04 | 0.66 | 0.04 | 0.54 | 0.03 | 0.46 | 0.12 |
| Cr# | 0.812 | 0.018 | 0.885 | 0.042 | 0.555 | 0.050 | 0.631 | 0.023 | 0.832 | 0.049 |
| Mg# | 0.579 | 0.041 | 0.382 | 0.041 | 0.667 | 0.022 | 0.538 | 0.026 | 0.466 | 0.125 |

### 4.2.1. Chromitite

The Mg#[Mg/(Mg + $Fe^{2+}$)] of unaltered chromitite spinels ranges between 0.51 and 0.64, and the Cr#[Cr/(Cr + Al)] ranges between 0.77 and 0.83. $TiO_2$ content ranges between 0.02 and 0.23 wt%. They can be classified as chromites (Figure 8a).

Chromite rims altered into ferrian chromite show lower Mg#, comprised between 0.31 and 0.45, and higher Cr#, comprised between 0.80 and 0.96.

Olivine, serpentine, and rare chlorite form the silicate gangue of Kymasi chromitites. The olivine Fo value ranges between 92.28 and 95.36%, and the NiO content ranges between 0.41 and 0.90 wt%. The Mg# of serpentine ranges between 0.93 and 0.96. Rare chlorite, associated with ferrian chromite alteration, shows high $Al_2O_3$ content (12.32–12.86 wt%) and $Cr_2O_3$ contents up to 3.45 wt%.

### 4.2.2. Partially Serpentinized Peridotite (F4d)

Spinels within partially serpentinized peridotite are aluminum chromites characterized by Mg# ranging between 0.63 and 0.66 and lower Cr# (0.55–0.62) with respect to chromitite spinels (Figure 8a). They are classified as Al-chromite.

Olivine is strongly forsteritic (Figure 8b), and shows Fo values ranging between 93.19 and 97.86%, and NiO content up to 0.66 wt%. Pyroxene is augitic, with a composition Wo 34%–42%, En 55%–63%, and Fs 2%–3% (Figure 8c). Serpentine (mainly lizardite) is not widespread, and shows Mg# of 0.97 and 1.00.

### 4.2.3. Dolomitized Peridotite (F4c)

Spinels within dolomitized peridotite are aluminum chromites, similar to spinels detected within partially serpentinized peridotite (Figure 8a), and show Mg# comprised between 0.51 and 0.54, and Cr# comprised between 0.58 and 0.66.

Primary silicates are enstatite (Figure 8c) with the following ranges: Wo 1.04%–2.23%, En 88.47%–90.85%, and Fs 7.97%–9.53%, and rare diopside relicts (Figure 8c) showing the following composition: Wo 46.79–48.34, En 49.30–50.10, and Fs 2.36%–3.28%. Serpentine shows Mg# between 0.85 and 0.94. Dolomite is widespread and shows the following compositional range: MgO (18.38–22.39 wt%) and CaO (28.23–31.19 wt%) (Figure 8d). Tremolite, present as a later alteration phase, has Mg# comprised between 0.97 and 0.99.

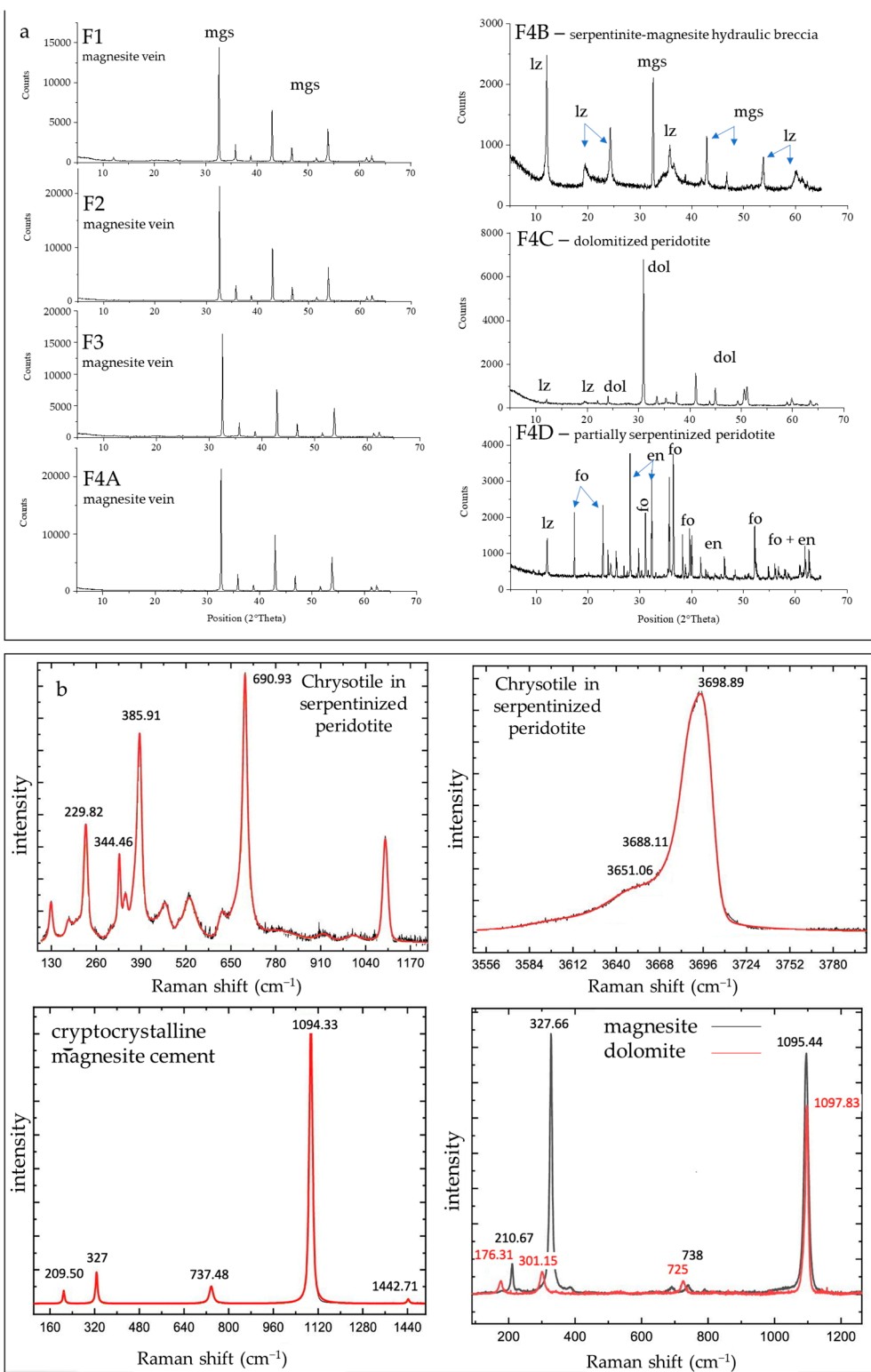

**Figure 7.** (**a**) XRPD pattern of magnesite samples (F1, F2, F3, F4a), serpentinite–magnesite hydraulic breccia (F4b), dolomitized peridotite (F4c), and partially serpentinized peridotite (F4d). Mgs: magnesite, lz: lizardite, dol: dolomite, fo: forsterite, en: enstatite. (**b**) Selected μ-Raman spectra of: chrysotile within a submillimetric veinlet in a partially serpentinized peridotite for the spectral region 110.86–1097.14 cm$^{-1}$ (**upper left**) and for the spectral region 3599.53–3698.89 cm$^{-1}$ (**upper right**); magnesite with massive cryptocrystalline texture within the cement of serpentinite–magnesite hydraulic breccia (**lower left**); magnesite and dolomite with microcrystalline polygonal texture within the cement of serpentinite–magnesite hydraulic breccia (**lower right**).

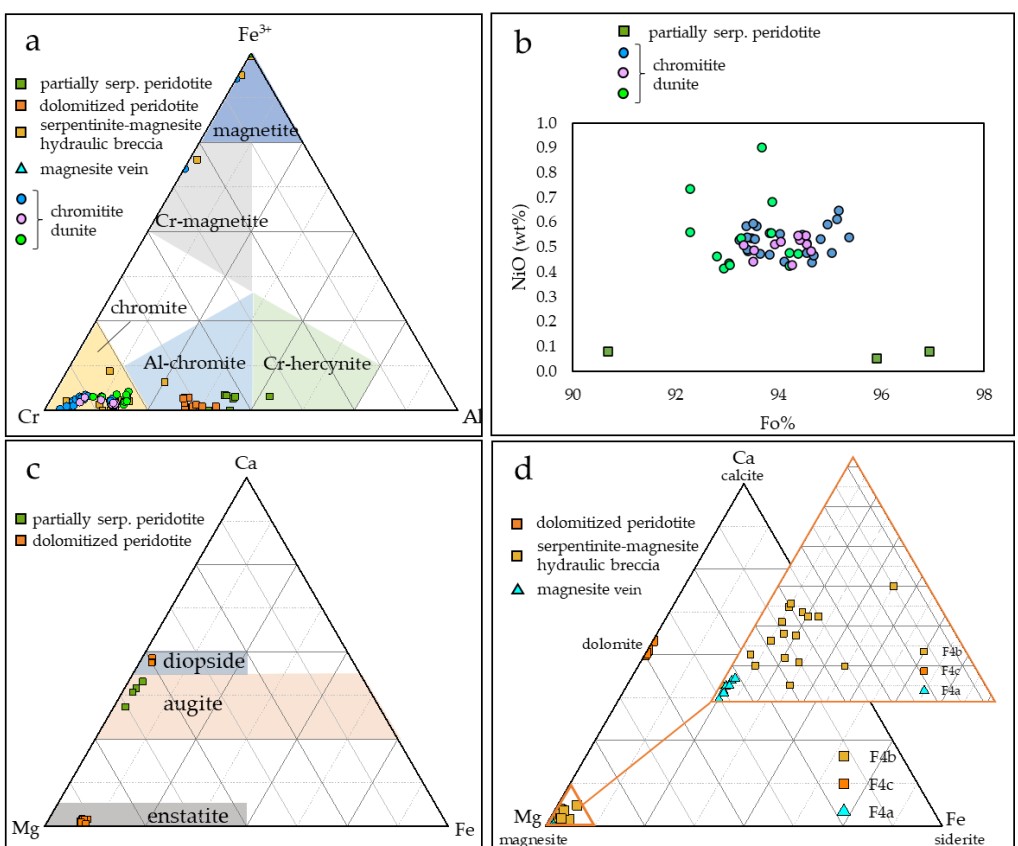

**Figure 8.** Ternary compositional diagrams of (**a**) spinels from chromitites and peridotites; compositional fields are from [24]; (**b**) olivine from chromitites and peridotites; (**c**) pyroxene from partially serpentinized peridotite and dolomitized peridotite; (**d**) carbonate phases from magnesite vein, serpentinite–magnesite hydraulic breccia and dolomitized peridotite.

### 4.2.4. Serpentinite–Magnesite Hydraulic Breccia (F4b)

Spinels within the serpentinite–magnesite breccia are chromites, with compositions more similar to those in chromitite than those in partially serpentinized peridotite and dolomitized peridotite (Figure 8a), with high Cr# (0.83–0.93) and Mg# comprised between 0.19 and 0.53.

Lizardite shows Mg# comprised between 0.91 and 0.98. Magnesite filling the fracture network shows MgO varying between 41.35 and 46.57 wt%, highly variable FeO, ranging from 0.12 to 4.07 wt%, and CaO varying between 0.35 and 4.01 wt%.

### 4.2.5. Magnesite Veins (F4a)

Magnesite filling the veins shows MgO comprised between 33.09 and 46.21 wt%, and low FeO and CaO comprised between 0.16 and 0.33 wt%, and 0.08 and 0.77 wt%, respectively.

### 4.3. Stable Isotope Geochemistry

Oxygen and carbon isotopic compositions of magnesite in magnesite veins within the serpentinite–magnesite hydraulic breccia, and of dolomite within the dolomitized peridotite are reported in Table 2. The $\delta^{13}C$ values are homogeneous within all magnesite samples, and vary between −10.63 and −12.68‰. $\delta^{18}O$ values (VSMOW) are also homogeneous among the different lithologies, varying between 26.33 and 27.72‰. Oxygen and carbon isotopic values of dolomite are comprised in the range of magnesite ones.

**Table 2.** $\delta^{13}C$ and $\delta^{18}O$ and standard deviation of carbonates from all lithologies.

| Sample | $\delta^{13}C$ (VPDB) | $\delta^{18}O$ (VPDB) | $\delta^{18}O$ (VSMOW) | St. Dev. $\delta^{13}C$ | St. Dev. $\delta^{18}O$ |
|---|---|---|---|---|---|
| F1 (mgs) | −12.68 | −3.24 | 27.56 | 0.03 | 0.04 |
| F2 (mgs) | −13.11 | −3.43 | 27.37 | 0.04 | 0.04 |
| F3 (mgs) | −10.63 | −3.09 | 27.72 | 0.04 | 0.03 |
| F4a (mgs) | −12.25 | −3.26 | 27.55 | 0.04 | 0.04 |
| F4b (mgs) | −12.53 | −4.02 | 26.76 | 0.03 | 0.06 |
| F4c (dol) | −11.74 | −3.46 | 27.33 | 0.04 | 0.06 |

## 5. Discussion

### 5.1. Chromitite and Peridotite Genesis

Magnesite veins at Kymasi are hosted within partially altered, chromitite-bearing peridotites. Spinels in the different lithologies show highly variable composition (Figure 9). Chromitites show high Cr# (>0.80) and a wide Mg# range (0.30–0.65). The serpentinite–magnesite hydraulic breccia, in contact with the magnesite vein, hosts spinels with a very similar mineral chemistry. On the contrary, the partially serpentinized peridotite and the dolomitized peridotite host spinels with lower Cr# (0.60–0.70) and narrow Mg# range (0.58–0.66).

Spinels within chromitites are quite homogenous, and are all high-Cr [25,26]. High-Cr chromitites are similar to other ophiolite chromite ores within ultramafic rocks in Greece, such as Vourinos [27–29] and partially Gomati [30,31], while they are quite different from chromitites from the Othris ophiolite [32] (Figure 9a). Disseminated spinels within peridotites at Kymasi are similar to spinels within Vourinos peridotites [27] and Gerakini serpentinites and harzburgites [11].

High-Cr chromitites generally indicate a genesis from high degrees of partial melting (~1200 °C) [33,34], from melts with boninitic affinity (Figure 9b,c). This is due to the fact that boninites commonly contain Cr-rich spinels, with a mineral chemistry similar to ophiolite ones [35], and are hence thought to be the parent melts of high-Cr ophiolite chromitites. Both Kymasi high-Cr chromitites and peridotites were formed in a supra-subduction forearc setting, as indicated by spinel mineral chemistry (Figure 9b). Chromitite ore bodies in supra-subduction settings are thought to be formed within "dunite channels", acting as pathways for the circulation of the parent melts of chromitites [36]. These dunite channels are generated by the dissolution of pyroxenes in supra-subduction zones, leaving a residual, porous dunite. A network of dunite channels provides the ideal conditions for the precipitation of chromitite bodies by the mixing of different melts [36,37].

Disseminated spinels in partially serpentinized peridotite and dolomitized peridotite have lower Cr# than spinels in chromitite, but, in any case, their composition reflects the one of dispersed spinels in restitic harzburgites [38]. On the contrary, disseminated spinels in the serpentinite–magnesite hydraulic breccia are high-Cr, with a mineral chemistry quite similar to the one in chromitites, and to spinels within dunite-hosting chromitites.

Chromites within ophiolites do not always preserve their magmatic imprint, due to sub-solidus re-equilibration, metamorphic, metasomatic, and hydrothermal processes during the obduction of the oceanic lithosphere [39]. Kymasi chromite grains show alteration into ferrian chromite at the rims (Figure 5a), while their cores are unaltered. The transformation of chromite into ferrian chromite involves the reaction of chromite and serpentine to form ferrian chromite and Cr-chlorite [39,40].

Chromite and olivine cores within densely disseminated chromitites are likely to retain their primary composition, and can be used as geothermometers, to estimate a primary temperature [28,41,42]. The major variables of olivine-spinel geothermometers are spinel and olivine Mg# (Mg/(Mg + $Fe^{2+}$), spinel Cr#*(Cr/Cr + Al + $Fe^{3+}$), $Fe^{3+}$# ($Fe^{3+}/\Sigma Fe$), and Ti moles [43,44] (Table 3). The major uncertainties are $Fe^{3+}$ and $Fe^{2+}$ calculations and the partial resetting of olivine-spinel compositions upon slow cooling at relatively high temperatures [41,45].

Primary temperatures at the core of chromite and olivine grains have been calculated on an average composition for three chromitite samples. They are similar for two samples

(780 and 788 °C) and higher for the third one (900 °C). As primary temperatures represent the temperature below which the compositional homogeneity within chromite and olivine cannot be maintained, it is always lower than the magmatic temperature of formation [42]. Kymasi primary temperatures are consistent with other chromitite temperatures in ophiolite environments, such as Vourinos (Greece) [27,28], Iballe (Albania) [38,42], Kluchevskoy (Russia) [46], and Luobusa (China) [47].

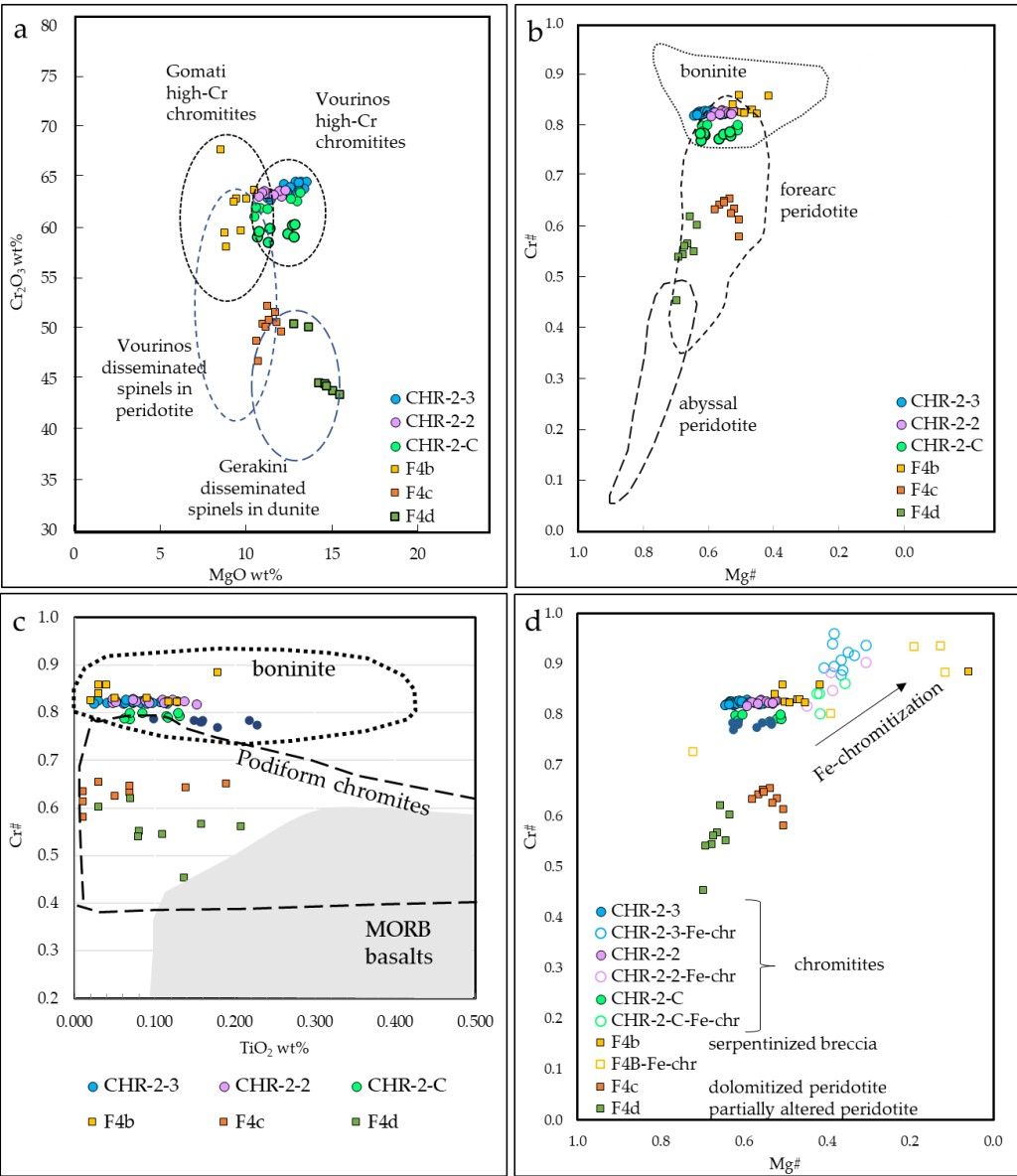

**Figure 9.** (**a**) $Cr_2O_3$ vs. MgO (wt%) of chromite cores. Compositional fields: Gomati high-Cr chromitites [30,31], Vourinos high-Cr chromitites [28], Vourinos disseminated spinels in dunite [27] Gerakini disseminated spinels in dunite [11]; (**b**) Cr# vs. Mg# of chromite cores. Compositional fields are from [35,38]; (**c**) Cr# vs. $TiO_2$ (wt%) of chromite cores. Compositional fields from [48]; (**d**) Cr# vs. Mg# of chromite cores and Fe-chromite rims.

## 5.2. Peridotite Carbonation and Magnesite Precipitation through Reactivation of Dunite Channels

In general, carbonation of ultramafic rocks is characterized by the replacement of pre-existing minerals by carbonate phases, mainly magnesite, dolomite, and calcite.

**Table 3.** Estimated primary temperatures and parameters for the olivine-spinel geothermometer according to Ballhaus [43]. The assumed pressure is 0.3 GPa.

| Parameters | CHR-2-3 Chromitite | CHR-2C Chromitite | CHR-2-2 Chromitite |
|---|---|---|---|
| Mg# olivine | 0.943 | 0.944 | 0.942 |
| Fe# olivine | 0.057 | 0.056 | 0.058 |
| Mg# spinel | 0.603 | 0.554 | 0.555 |
| Fe# spinel | 0.397 | 0.446 | 0.445 |
| Cr*# spinel | 0.823 | 0.794 | 0.824 |
| $Fe^{3+}$# spinel | 0.122 | 0.096 | 0.059 |
| Ti moles | 0.002 | 0.002 | 0.002 |
| KD | 10.919 | 13.481 | 13.071 |
| T (°C) | 900 | 788 | 781 |

At Kymasi, carbonation led to the formation of a thin centimetric layer of dolomitized peridotite close to massive magnesite veins. Magnesite at Kymasi occurs within serpentinite–magnesite hydraulic breccia, as well as within magnesite-filled fractures. Carbonate precipitation is caused by the interaction of serpentinized peridotites with $CO_2$-rich fluids, usually derived from sea water and the atmosphere [49,50].

Carbonation of ultramafic rocks and circulation of $CO_2$-rich fluids at Kymasi resulted in the replacement of pyroxene and possibly olivine and serpentine by dolomite within the host peridotite, and in the precipitation of massive magnesite in veins. Progressing from the partially serpentinized peridotite to the magnesite veins, we can find a dolomitized peridotite and a serpentinite–magnesite hydraulic breccia. Dolomite is the main mineralogical phase within the dolomitized peridotite. Its formation can be linked to the alteration of Ca-Mg bearing silicates, such as tremolite or clinopyroxene. Previous studies showed that peridotite serpentinization is characterized by the hydration of olivine and pyroxene, producing serpentinites with mesh textures and bastites, respectively. The excess of $Ca^{2+}$ and $Mg^{2+}$ from the alteration of primary silicates can react with $CO_2$ to produce calcite or dolomite [51]. Spinel mineral chemistry of dolomitized peridotite is consistent with the nearby partially serpentinized peridotite, with relatively low-Cr spinels.

The serpentinite–magnesite hydraulic breccia, closer to magnesite veins, records a brittle event. Serpentine clasts are angular, and cemented by magnesite with a cockade texture. The low amount of dolomite and the spinel mineral chemistry, more similar to Kymasi chromitites, suggests that carbonation affected a dunite.

The fluids responsible for carbonation and precipitation of magnesites circulated through preferential pathways, possibly reactivating older weakness zones. Chromitite-hosting dunites are formed by percolation of an ascending melt through mantle peridotite in channels where high degree of pyroxene melting weakens the peridotite (Figure 10a), leaving a restitic dunite [36,37,52,53]. Chromitite ores may form in these weakness zones by mixing of two different melts (Figure 10b).

Spinels show compositions similar to those of Kymasi chromitites that cannot be explained by any post-magmatic re-equilibration in spinel composition as post-magmatic processes were limited to minor ferrian-chromitization at rims. The similarity can hence be attributed to the dunitic nature of the carbonated rock.

During obduction, ophiolites undergo strong deformation in a fragile regime, where these weakness zones can be reactivated as faults or shear zones that work as pathways for the circulation of $CO_2$-rich fluids, precipitating the typical stockwork magnesite veins (Figure 10c). The system of veins sampled at Kymasi exploits an old dunite channel, as suggested by spinel mineral chemistry, and the circulating fluids partially affect also the hosting peridotite, in the specific case inducing dolomitization of the host peridotite (Figure 10d).

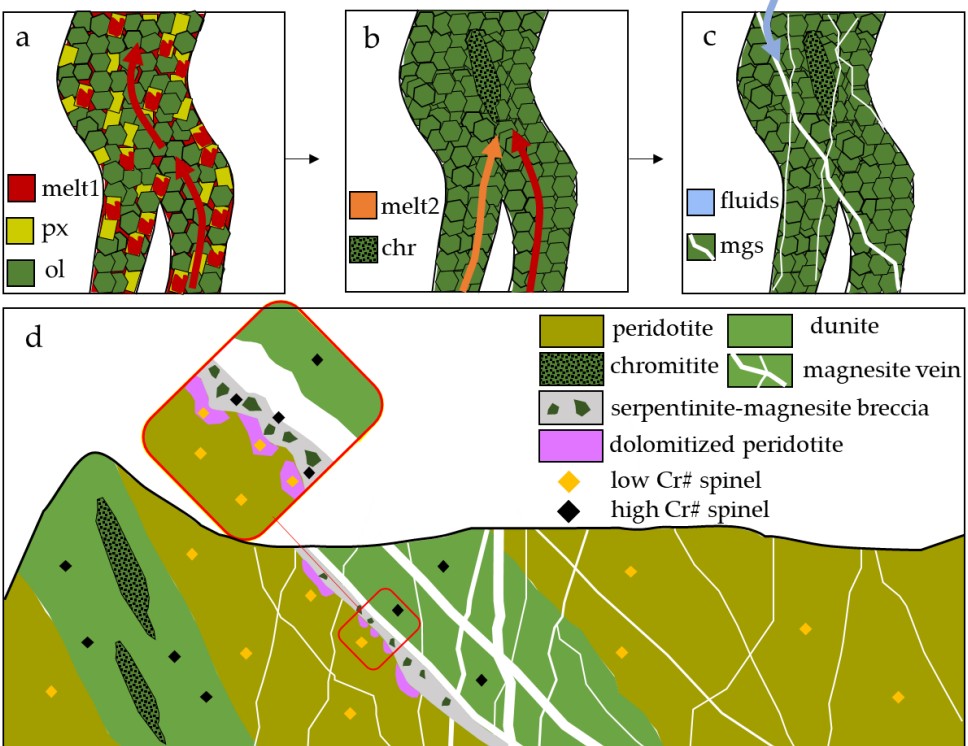

**Figure 10.** Schematic genetic model showing (**a**) infiltration of melts ascending through peridotite and inducing pyroxene melting, and the formation of a restitic dunite; (**b**) different melts can mix and induce precipitation of chromitite due to chromite oversaturation; (**c**) after the obduction of the oceanic crust, the weaker zones of the dunite channels can be reactivated as $CO_2$-rich fluid pathways, precipitating magnesite in stockwork veins; (**d**) out-of-scale scheme of Kymasi chromitites and magnesite main veins, showing the difference in spinel mineral chemistry.

### 5.3. Origin of Magnesite-Forming Fluids

The low-T reaction between surface water and peridotite produces $Mg$-$HCO_3^-$ waters, also called Type-1 waters [5,6,54–56]. Magnesite, dolomite, serpentine, and clay then precipitate from the reaction between peridotites and $Mg$-$HCO_3^-$ waters, once they become isolated from the atmosphere [6,55–58]. The resulting waters are enriched in Ca (from the alteration of pyroxene) and $OH^-$, and depleted in dissolved carbon [5]. These Ca-$OH^-$-rich waters are called Type 2 waters [5,9], and generally form alkaline springs emerging from peridotites. Finally, the alkaline waters can react with the atmosphere to precipitate calcite [9].

Carbon and oxygen stable isotope analyses of magnesite veins are crucial for understanding the origin and source of the mineralizing fluids. The narrow range of $\delta^{18}O$ and $\delta^{13}C$ isotopes, common to other ultramafic-hosted magnesite deposits in the Balkans [59] (Figure 11), suggests that the mineralization was formed through a single process.

$\delta^{18}O$ values of Kymasi magnesite veins, comprised between 26.76 and 27.72 ‰, are comparable to the values of magnesite veins of the California Coast Ranges [60,61] and the Oman magnesite deposits [9], indicating an equilibrium with local waters between 15 and 40 °C. These oxygen isotopic ratios reflect those of meteoric waters, implying a low-T precipitation of the magnesite mineralization very close to the surface.

$\delta^{13}C$ isotopic values can help identify the source of the carbon. Kymasi negative values, comprised between −10 and −15, rule out the possibility of a deep-seated or mantle source of the $CO_2$ [59,62], which would produce higher isotopic ratios. $\delta^{13}C$ values are more similar to other magnesite ores formed in shallow environments [59,63] and from a biogenic source of the carbon [64]. The organic source of carbon can be provided either through decarboxylation of organic sediments (∼−15‰), by thermal contact metamor-

phism decomposition of limestone [7,59,65], by chemical weathering with involvement of atmospheric $CO_2$ (~3‰) [4,7,64,66], or by a mixed source. Strongly negative $\delta^{13}$C PDB values, in any case, point towards an organic carbon source.

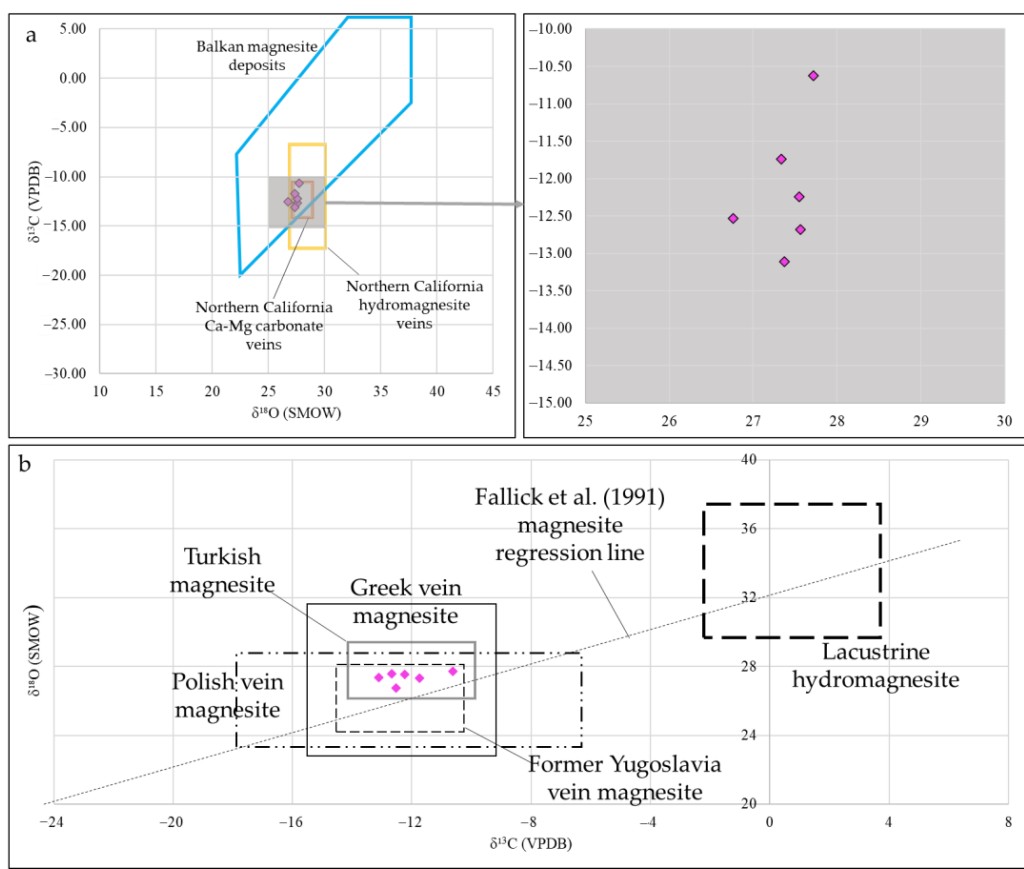

**Figure 11.** $\delta^{13}$C vs. $\delta^{18}$O (**a**) and $\delta^{18}$O vs. $\delta^{13}$C (**b**) of Kymasi magnesite. Compositional fields are: Balkan magnesite deposits [59], Northern California hydromagnesite veins [9], Northern California Ca-Mg carbonate veins [9], Turkish magnesite [65], Polish vein magnesite [64], Greek vein magnesite [67], former Yugoslavian vein magnesite, and lacustrine hydromagnesite [59].

## 6. Conclusions

The coexistence of dunite-hosted chromitite lenses and magnesite vein stockwork within the Kymasi ophiolite mantle offers the opportunity to unravel the sequence of events that affected the rocks during a long time-span, with changes in geotectonic settings: from old deep mantle metasomatism, down to recent carbonation. The following steps can be reconstructed:

Percolation of fluid-rich Cr-bearing melts in a supra-subduction mantle wedge with the formation of dunite channels hosting chromitite lenses;

Obduction of the ophiolite sequence, and partial serpentinization of peridotites;

Uplift and erosion of mantle rocks to a shallow level in the crust;

Percolation at low temperature of meteoric waters rich in biogenic carbon into the mantle rocks at shallow depth, reactivating the dunite channels as preferential weak zones;

Precipitation of magnesite in veins and partial carbonation of peridotite host rock;

Further erosion till exposure at the surface.

The close spatial association of chromitite and magnesite ores in the Gerakini and Vavdos ophiolite mantle rocks of the Halkidiki peninsula envisages the possibility that the genetic relationship between these two kinds of ores, due to their formation within the same weakness zones active in two different dynamic regimes, can be not just a peculiarity of Evia ophiolites, but a more widely occurring process.

**Supplementary Materials:** The following supporting information can be downloaded at: https://www.mdpi.com/article/10.3390/min13020159/s1, Table S1: sil-chromitite; Table S2: sil-carb-peridotite; Table S3: spi-nel-chromitite; Table S4: spinel-peridotite.

**Author Contributions:** Conceptualization, G.G. and M.B.; methodology, M.B., P.M., L.C. and E.T.; validation, A.C., P.M. and L.C.; formal analysis, M.B.; investigation, M.B. and G.G.; resources, G.G. and A.C.; data curation, M.B. and G.G.; writing—original draft preparation, M.B.; writing—review and editing, G.G.; visualization, M.B.; supervision, G.G.; project administration, G.G.; funding acquisition, G.G. All authors have read and agreed to the published version of the manuscript.

**Funding:** This research was funded by the Italian Ministry of Education (MUR) through the project "PRIN2017—Mineral reactivity, a key to understand large-scale processes" and the project "Dipartimenti di Eccellenza 2017".

**Data Availability Statement:** All data are available and can be found in the article and in the Supplementary Materials.

**Acknowledgments:** The authors wish to acknowledge Nicola Campomenosi for technical support in the μ-Raman analyses. We also wish to acknowledge MSc students Irene Albieri, Simone Grasso, and Matteo Valentini for their work in the field. We also wish to thank the reviewers for their constructive comments.

**Conflicts of Interest:** The authors declare no conflict of interest.

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
