# Peer review of "The Formation of Magnesite Ores by Reactivation of Dunite Channels as a Key to Their Spatial Association to Chromite Ores in Ophiolites: An Example from Northern Evia, Greece"

_minerals, doi:10.3390/min13020159_

Round 1

Reviewer 1 Report

This paper presents a short study of the formation of magnesite in dunites and serpentinised peridotites from a locality in Greece. It show based on stable isotopes that magnesite was formed at low temperature after ophiolite emplacement by meteoric water with organic carbon, and that veins used zones of weaknesses (dunite channels) to penetrate the peridotites.

The paper is well written and nicely presented, and I don’t have much criticism, mainly some small questions and comments.

One minor point which could use a little more discussion is why dunites are zone of weakness and thus pathways for fluids. The authors make the observation but don’t really provide an explanation.

The other point is that perhaps the authors could mention CO2 sequestration. These rocks appear to be a useful natural analogue.

I recommend publication after minor revision taking into account my comments below.

l.47 ‘serpentine is highly soluble in pure water’ with a somewhat obscure reference. It is clearly not, no silicates are highly soluble. Also, in terms of the model, it is more likely that Mg fluids are produced during serpentinisation, when olivine breaks down, and serpentine is formed.

l. 55 present case = present study

l. 63 exposition = exposure

Fig. 1 (a) Box missing for Internal Hellenides in legend. (b) bottom right area look strange, what’s the white area?

Fig. 2 Ideally GPS coordinates of this locality would be presented.

l. 136 beam diameter is 1 µm (µ missing)

4.1.1. What are olivine Fo and TiO2 in chromite? These are useful for interpreting petrogenesis.

l. 169 No carbonate minerals were detected. You mean, other than magnesite?

Fig. 6. Would be useful to indicate which samples the photographs are from.

l. 221 Clarify here or in the table with the data how Fe2O3 and FeO are determined since EPMA does not discriminate between those (I assume it’s calculated based on stoichiometry)

4.2 and subsections. Presenting MgO and FeO repeatedly for different minerals in different rock types isn’t very useful, we can see this in the tables. Please present Fo for olivne, Mg# for other minerals. For Cpx it would be useful to know WoEnFs composition.

Fig. 8a. Is there a reference for the subdivisions? Pickotite is not a name I have seen before.

Fig. 8b. This olivine diagram is not very useful, as Ca is not an olivine endmember. It would be much more instructive to provide a X-Y plot of Fo content vs NiO, not a ternary diagram.

Table 1. Since you give standard deviations, please also provide number of analyses (n).

l. 326-340 ‘likely retain their magmatic composition’ This is simply not true, and you say that yourself a few lines later: olivine-spinel compositions re-equilibrate to lower temperatures due to Mg-Fe exchange. This is very common if not unavoidable in ophiolites, see e.g., Chen et al. 2020 (https://doi.org/10.1016/j.lithos.2020.105773). What you get are cooling temperatures, so the temperatures are dependent on how quickly the rocks cooled during or after emplacement of the ophiolite, as well as on the grain size of spinel. In contrast (to Fe and Mg), Cr, Al and TiO2 in spinel/chromite are preserved (as they are very low in olivine) so they are useful for reconstructing their origin. This needs a little bit of rephrasing.

Table 3. R is a constant, no need to put in table. And P can be mentioned in caption.

Fig. 9C I don’t know the reference for the fields in the Cr#-TiO2 diagram, but surely it’s incorrect as MOR peridotite does not contain 0.3 – 1.0 wt% TiO2. If they do, they’re melt impregnated. See recent paper by Whattam et al. (https://doi.org/10.1007/s00410-022-01968-9) which presents MOR spinel data, the majority being <0.2 wt TiO2. Unless it’s the field for MOR lavas, not peridotite?

The division between depleted and highly depleted also seems arbitrary, how are these defined?

l. 375 But why would the dunites be zones of weakness? Is there any reason why they would be more permeable to fluids at low temperature?

l. 442 It would be useful if you briefly explained how the two ore-forming processes are connected.

Author Response

REVIEWER 1

One minor point which could use a little more discussion is why dunites are zone of weakness and thus pathways for fluids. The authors make the observation but don’t really provide an explanation.

We agree this point required some improvement, so we implemented the discussion about weakness zones following specific suggestion of the reviewers and editor below.

The other point is that perhaps the authors could mention CO2 sequestration. These rocks appear to be a useful natural analogue. We added a brief paragraph in the introduction, citing also natural magnesite as a natural case study to better understand the rates of mineral carbonation and CO2 sequestration.

l.47 ‘serpentine is highly soluble in pure water’ with a somewhat obscure reference. It is clearly not, no silicates are highly soluble. Also, in terms of the model, it is more likely that Mg fluids are produced during serpentinisation, when olivine breaks down, and serpentine is formed. Yes, we are thankful for pointing that out, we changed the sentence accordingly.

  1. 55 present case = present study ok
  2. 63 exposition = exposure ok

Fig. 1 (a) Box missing for Internal Hellenides in legend. (b) bottom right area look strange, what’s the white area? (a) the box was not missing, we just decided to expand the Internal Hellenides and put all the lithologies. To clarify it, we put all the lithologies belonging to the Internal Hellenides in a box. (b) We changed the color and added it in the legend.

Fig. 2 Ideally GPS coordinates of this locality would be presented. We added the coordinates in the text

  1. 136 beam diameter is 1 µm (µ missing) ok

4.1.1. What are olivine Fo and TiO2 in chromite? These are useful for interpreting petrogenesis. We added the Fo content of olivine and the TiO2 content in the mineral chemistry paragraph, as suggested in the comment.

  1. 169 No carbonate minerals were detected. You mean, other than magnesite? We mean that in the serpentinized host rock there is no carbonate phase (including magnesite). Of course it hosts big magnesite veins.

Fig. 6. Would be useful to indicate which samples the photographs are from. Ok

  1. 221 Clarify here or in the table with the data how Fe2O3 and FeO are determined since EPMA does not discriminate between those (I assume it’s calculated based on stoichiometry). Yes, it is calculated based on stoichiometry, we added it in the table caption.

4.2 and subsections. Presenting MgO and FeO repeatedly for different minerals in different rock types isn’t very useful, we can see this in the tables. Please present Fo for olivne, Mg# for other minerals. For Cpx it would be useful to know WoEnFs composition. Ok

Fig. 8a. Is there a reference for the subdivisions? Pickotite is not a name I have seen before. Picotite is a term used in the past to refer to a variety of hercynite. As it is not used anymore, we changed the diagram to include only known terms, and added the new reference.

Fig. 8b. This olivine diagram is not very useful, as Ca is not an olivine endmember. It would be much more instructive to provide a X-Y plot of Fo content vs NiO, not a ternary diagram. Yes, we agree, we replaced the ternary diagram with a Fo-NiO one.

Table 1. Since you give standard deviations, please also provide number of analyses (n). Ok.

  1. 326-340 ‘likely retain their magmatic composition’ This is simply not true, and you say that yourself a few lines later: olivine-spinel compositions re-equilibrate to lower temperatures due to Mg-Fe exchange. This is very common if not unavoidable in ophiolites, see e.g., Chen et al. 2020 (https://doi.org/10.1016/j.lithos.2020.105773). What you get are cooling temperatures, so the temperatures are dependent on how quickly the rocks cooled during or after emplacement of the ophiolite, as well as on the grain size of spinel. In contrast (to Fe and Mg), Cr, Al and TiO2 in spinel/chromite are preserved (as they are very low in olivine) so they are useful for reconstructing their origin. This needs a little bit of rephrasing.

Yes we agree, both olivine and spinel re-equilibrate to lower temperatures due to Fe-Mg exchange, but the effect of re-equilibration on the grains depends on size of the crystals and volume ratio between olivine and chromite. In massive chromitites chromite grains are affected by re-equilibration only close to the rim, while the core area shows homogeneous primary composition, as reported by the articles cited in the text.

We anyway changed the term magmatic, which may be misleading, with the more general “primary”.

Table 3. R is a constant, no need to put in table. And P can be mentioned in caption. Ok

Fig. 9C I don’t know the reference for the fields in the Cr#-TiO2 diagram, but surely it’s incorrect as MOR peridotite does not contain 0.3 – 1.0 wt% TiO2. If they do, they’re melt impregnated. See recent paper by Whattam et al. (https://doi.org/10.1007/s00410-022-01968-9) which presents MOR spinel data, the majority being <0.2 wt TiO2. Unless it’s the field for MOR lavas, not peridotite?The division between depleted and highly depleted also seems arbitrary, how are these defined?

We decided to change the diagram to plot more significative compositional fields. We maintained boninites but we added podiform chromites and mid ocean ridge basalts compositional fields (from Pagé and Barnes 2009). We hope that now it is clearer.

  1. 375 But why would the dunites be zones of weakness? Is there any reason why they would be more permeable to fluids at low temperature?

We understand that this text required some clarification, and we changed it in order to better explain this point. Dunites are not weakness zones but, according to references cited in the text, they form in weakness zones in ductile regime that can be reactivated during obduction in a fragile regime.

  1. 442 It would be useful if you briefly explained how the two ore-forming processes are connected.

We think that the change of text at line 375 can also better clarify the connection of the two ore forming processes. They are probably related to the same weakness zones active in two different dynamic regimes. Anyway we added one more sentence about this relation in the conclusions.

Reviewer 2 Report

Dear Authors, 

I found the following problems in your manuscript: 

1. The color and thickness of the arrows in Figures 1 and 6 should be changed. Now they are too narrow to spot them, and also the white color is not helping with that. 

2. If possible, I suggest using a better-quality picture for figure 3. Now the figure is blurred. 

3. Unfortunately, the methodology description of XRD is lacking important data. I would appreciate it if the authors may add the following information to the XRD description: 

 - Which method of PXRD was used? It was the Bragg-Brentano method or DSH? 

- Time of data acquisition. It can be presented as a step size 

4. Similar story is with an SEM-EDS methodology. It needs to be rewritten and include the information: 

- Coating: Material used for coating and its thickness. Which coating device was used and what parameters of the coating were used? 

- SEM parameters: Acceleration voltage, spot size (/current), if the measurements were done without the coating in eg. low vacuum mode, etc.  

- The manufacturer and model of the EDS system 

5. In EPMA methodology is lacing the thickness of the coating. I assume it is 20 nm of C, but I will appreciate it if the authors may add this information to the methodology section. 

6. In the manuscript I spotted that authors write the name of the samples in different ways eg. somewhere is written "F4c" and in other parts of the manuscript is "F4C". I would suggest using only one style of writing. Now, it is a little bit confusing. 

7. I think that paragraph 4.1 should be deleted or rewritten. Its current form does not bring anything to the manuscript. It should contain some figures which may be published as supplementary material. It can be also shifted somewhere in the manuscript because I notice that figure 7 have information that is described in this section. Or figure 7 may be divided into XRD and Raman figures, where the XRD figure should be put in this paragraph. 

I am looking forward to receiving your manuscript after corrections.  

Author Response

REVIEWER 2

  1. The color and thickness of the arrows in Figures 1 and 6 should be changed. Now they are too narrow to spot them, and also the white color is not helping with that. We changed the color and the thickness of the lines in the figures.
  2. If possible, I suggest using a better-quality picture for figure 3. Now the figure is blurred. Unfortunately, that is the best picture we have of the outcrop in which the field relationship between the lithologies can be seen.
  3. Unfortunately, the methodology description of XRD is lacking important data. I would appreciate it if the authors may add the following information to the XRD description: 

 - Which method of PXRD was used? It was the Bragg-Brentano method or DSH? We added this information.

- Time of data acquisition. It can be presented as a step size We added this information

  1. Similar story is with an SEM-EDS methodology. It needs to be rewritten and include the information: 

- Coating: Material used for coating and its thickness. Which coating device was used and what parameters of the coating were used?  We added this information.

- SEM parameters: Acceleration voltage, spot size (/current), if the measurements were done without the coating in eg. low vacuum mode, etc.  We added this information.

- The manufacturer and model of the EDS system We added this information.

  1. In EPMA methodology is lacing the thickness of the coating. I assume it is 20 nm of C, but I will appreciate it if the authors may add this information to the methodology section.  We added this information.
  2. In the manuscript I spotted that authors write the name of the samples in different ways eg. somewhere is written "F4c" and in other parts of the manuscript is "F4C". I would suggest using only one style of writing. Now, it is a little bit confusing. We homogenized the name of the samples in the text and in the Figures.
  3. I think that paragraph 4.1 should be deleted or rewritten. Its current form does not bring anything to the manuscript. It should contain some figures which may be published as supplementary material. It can be also shifted somewhere in the manuscript because I notice that figure 7 have information that is described in this section. Or figure 7 may be divided into XRD and Raman figures, where the XRD figure should be put in this paragraph. 

We received contrasting reviews about paragraph 4.1 and its figures. We believe that the mineralogy and texture of the samples is essential to the comprehension of the paper, and therefore we decided not to delete it. About Figure 7, both XRD and Raman provide useful information for the mineralogy, that is complementary to the optical microscopy description and essential for a thorough understanding of the mineral assemblage.

Round 2

Reviewer 2 Report

The authors corrected all issues and response to all my comments. I do not have any more suggestions. 

Author Response

-